# Oncolytic Vaccinia Virus Augments T Cell Factor 1-Positive Stem-like CD8^+^ T Cells, Which Underlies the Efficacy of Anti-PD-1 Combination Immunotherapy

**DOI:** 10.3390/biomedicines10040805

**Published:** 2022-03-30

**Authors:** Yun-Hui Jeon, Namhee Lee, Jiyoon Yoo, Solchan Won, Suk-kyung Shin, Kyu-Hwan Kim, Jun-Gyu Park, Min-Gang Kim, Hang-Rae Kim, Keunhee Oh, Dong-Sup Lee

**Affiliations:** 1Department of Biomedical Sciences, Seoul National University College of Medicine, Seoul 03080, Koreascwon@snu.ac.kr (S.W.); skshin@snu.ac.kr (S.-k.S.); kyuhwan1992@gmail.com (K.-H.K.); gozjsh@gmail.com (J.-G.P.); mingang.kim@snu.ac.kr (M.-G.K.); hangrae2@snu.ac.kr (H.-R.K.); 2Wide River Institute of Immunology, Seoul National University, Hongcheon-gun 25159, Korea; 3SillaJen, Inc. Research Center, Seongnam-si 13493, Korea; nhlee@kr.sillajen.com

**Keywords:** oncolytic vaccinia virus, cancer antigen-specific T cells, stem-like CD8^+^ T cells, lymphoid organs, immune checkpoint blockade

## Abstract

Oncolytic virotherapy has garnered attention as an antigen-agnostic therapeutic cancer vaccine that induces cancer-specific T cell responses without additional antigen loading. As anticancer immune responses are compromised by a lack of antigenicity and chronic immunosuppressive microenvironments, an effective immuno-oncology modality that converts cold tumors into hot tumors is crucial. To evaluate the immune-activating characteristics of oncolytic vaccinia virus (VACV; JX-594, pexastimogene devacirepvec), diverse murine syngeneic cancer models with different tissue types and immune microenvironments were used. Intratumorally administered mJX-594, a murine variant of JX-594, potently increased CD8^+^ T cells, including antigen-specific cancer CD8^+^ T cells, and decreased immunosuppressive cells irrespective of tissue type or therapeutic efficacy. Remodeling of tumors into inflamed ones by mJX-594 led to a response to combined anti-PD-1 treatment, but not to mJX-594 or anti-PD-1 monotherapy. mJX-594 treatment increased T cell factor 1-positive stem-like T cells among cancer-specific CD8^+^ T cells, and anti-PD-1 combination treatment further increased proliferation of these cells, which was important for therapeutic efficacy. The presence of functional cancer-specific CD8^+^ T cells in the spleen and bone marrow for an extended period, which proliferated upon encountering cancer antigen-loaded splenic dendritic cells, further indicated that long-term durable anticancer immunity was elicited by oncolytic VACV.

## 1. Introduction

The efficacy of oncolytic virotherapy for cancer was originally thought to be dependent on selective lysis of infected cancer cells; therefore, efficient cancer targeting and extensive intratumoral virus replication and propagation were considered limiting factors for cancer treatment [1]. However, subsequent studies have shown that oncolytic viruses (OVs) effectively induce immunogenic cancer cell death and release tumor-associated antigens (TAAs), suggesting that OVs evoke systemic immune responses against cancer antigens [2,3]. Therefore, OVs have been proposed as ideal agents for activating cancer-specific T cell responses, converting them into immunologically hot tumors, which is a prerequisite for the currently prevailing immune checkpoint blockade (ICB) therapies, such as anti-programmed cell death protein 1 (PD-1)-blocking antibodies, and eventually synergizing them with these antibodies [4,5]. Indeed, oncolytic herpes simplex virus, which was the first United States Food and Drug Administration-approved treatment for advanced melanoma, showed synergistic anti-cancer activity with anti-PD-1 immunotherapy by promoting intratumoral T cell infiltration [6].

Immune responses against cancer are hampered by the poor activation of cancer-specific T cells and the suppressive nature of the tumor immune microenvironment [7,8]. Immune evasion by cancer results in a persistent antigen presence leads to exhausted T cell phenotypes in cancer [9]. These cells are divided into weakly and terminally exhausted T cells, where the former can be reinvigorated by ICB treatment [10]. Stem-like T cells show proliferative capacity following ICB treatment, and T cell factor 1 (TCF1) is an important marker for these cells [11]. A high percentage of stem-like T cells in the tumor is correlated with a good cancer prognosis [12] and a robust response to ICB [13]. The intratumoral niche has been proposed to be involved in the maintenance and differentiation of stem-like CD8^+^ T cells [14], and tumor-draining lymph nodes act as the reservoir for ongoing antitumor immune responses [15]. In addition, activation of stem-like CD8^+^ T cells is important for the therapeutic efficacy of nanoparticle vaccinations [16] and adoptive cell immunotherapy [17]. OVs have been considered antigen-agnostic cancer vaccines, as they kill cancer cells, release TAA, and induce TAA-specific T cell responses even without additional neo-antigens [18]. Thus, we consider stem-like cancer antigen-specific CD8^+^ T cells to be key components of the OV-mediated immune response and synergistic action of combination therapies involving ICB.

Vaccinia virus (VACV) is highly immunogenic and has a broad range of host cells; therefore, it is an ideal candidate for oncolytic virotherapy [19]. Pexa-vec (pexastimogene devacirepvec, JX-594) is an oncolytic VACV with a defective viral thymidine kinase (TK) gene. It can activate dendritic cells (DCs) via granulocyte-macrophage colony-stimulating factor (GM-CSF) and has exerted marked antitumor effects (with low toxicity) through oncolytic, anti-angiogenic, and immune-stimulating activities in preclinical and clinical studies [20]. Recent studies have shown that a murine variant of JX-594 (mJX-594; WR.TK^−^ murine GM-CSF) robustly modulates the tumor microenvironment (TME), thereby increasing the efficacy of ICB in kidney cancer and peritoneal carcinomatosis of colon cancer [21,22,23].

Here, we comprehensively evaluated immune microenvironment changes in murine syngeneic cancer models following mJX-594 treatment and found that mJX-594 potently increased CD8^+^ T cells, including cancer antigen-specific CD8^+^ T cells, and decreased immunosuppressive cells in the tumors irrespective of therapeutic efficacy. Remodeling of tumors into inflamed ones by JX-594 led to a response to combined anti-PD-1 treatment in tumors, where single therapy, of either high-dose JX-594 or anti-PD-1 antibody, did not have significant tumor-reducing effects. mJX-594 treatment increased effector T cell and TCF1^+^ stem-like T cell expression among cancer-specific CD8^+^ T cells in the tumor tissue and spleen; anti-PD-1 antibody further increased these cells, which was important for therapeutic efficacy. These stem-like CD8^+^ T cells survived for an extended period in the secondary lymphoid organs and bone marrow and were functional in terms of in situ proliferation by antigen-loaded DCs, further demonstrating the long-term anticancer immunity induced by immunotherapy with oncolytic VACV and ICB.

## 2. Materials and Methods

### 2.1. Experimental Animals

C57BL/6 and BALB/c mice aged 7–8 weeks were purchased from Koatech (Pyeongtaek, Korea) and housed under standard specific pathogen-free conditions at Seoul National University Animal Facility (Seoul, Korea). All animal experiments were performed with the approval of the Institutional Animal Care and Use Committee of Seoul National University Hospital (no.19-0107-S1A1, 25 November 2019).

### 2.2. Cell Lines

EMT6 (experimental mammary tumour-6, murine triple-negative breast cancer cells of BALB/c origin), B16-F10 (B16 melanoma F10, murine melanoma cancer cells of C57BL/6 origin), RENCA (renal carcinoma, murine renal cell carcinoma cells of BALB/c origin), LLC (Lewis lung carcinoma, murine lung cancer cells of C57BL/6 origin), MB49 (mouse bladder-49, murine bladder cancer cells of BALB/c origin), MC38 (murine carcinoma-38, murine colon cancer cells of C57BL/6 origin), CT26 (colon tumor-26, murine colon cancer cells of BALB/c origin), and 4T1 (murine metastatic triple-negative breast cancer cells of BALB/c origin) cancer cell lines were obtained from the American Type Culture Collection (ATCC) (Manassas, VA, USA). As a cancer-specific immune response model, we established an LLC cell line expressing ovalbumin (OVA) (LLC-OVA) by transfecting the gene encoding cytosolic whole OVA protein into the LLC cell line using a lentiviral expression vector. Cancer cell lines were maintained in a complete culture medium consisting of either RPMI 1640 or Dulbecco’s modified Eagle’s medium (DMEM), 10% *v*/*v* fetal bovine serum (FBS; Gibco-BRL, Gaithersburg, MD, USA), and 1% *v*/*v* antibiotic/antimycotic under sterile conditions at 37 °C in a 5% *v*/*v* CO_2_ atmosphere.

### 2.3. Oncolytic Virus

mJX-594, a mouse variant of JX-594, was propagated and provided by SillaJen, Inc. (Seoul, Korea). mJX-594 is a Western Reserve strain of VACV encoding murine GM-CSF in the vaccinia TK gene locus under the control of the p7.5 promoter. To amplify the virus, the host cell HeLa was infected with the virus at 0.02 MOI (multiplicity of infection) for 48 h and the infected cells were lysed in hypotonic lysis buffer. The infected cell lysate was filtered to eliminate host cell debris and concentrated by 36% sucrose cushion centrifugation. The virus was stored at −80 °C.

### 2.4. Tumor Models and Treatment Regimens

To generate tumor models, 2 × 10^5^ cancer cells were implanted by subcutaneous injection into the right flank fat pads of wild-type C57BL/6 and BALB/c mice. When the tumor volumes reached 50–60 mm^3^, mice with size-matched tumors were randomly assigned to the experimental groups, followed by intratumoral injection of either vehicle (phosphate-buffered saline, PBS), 1 × 10^7^, or 5 × 10^7^ plaque-forming units (pfu) of mJX-594, three or four times at 3-day intervals. Vehicle and virus were prepared in a volume of 40 μL per tumor burden for one mouse. Tumor growth was monitored every 2–3 days until the end of the experiment (tumor volume ≤ 1500 mm^3^). Tumor volumes were calculated as (width × width × length)/2.

To assess the proliferative activity of cancer antigen-specific memory CD8^+^ T cells, LLC-OVA cancer-bearing mice were intratumorally treated with either vehicle or 5 × 10^7^ pfu mJX-594 on days 0, 3, and 6, and primary tumors were resected 25 days after the first mJX-594 treatment to improve survival. Three days before analysis, 3 × 10^6^ splenic DCs loaded with either vehicle or OVA 357–364 peptide (SIINFEKL) were injected intravenously into the mice. Thereafter, the mice were twice administered 10 mg/kg of EdU (5-ethynyl-2′-deoxyuridine; Invitrogen, Carlsbad, CA, USA) on 2 consecutive days. DCs were prepared by incubation with 100 nM peptide for 2 h at 37 °C just prior to injection. Twenty-four hours after the second EdU labeling, EdU incorporation was measured by staining the cells isolated from the mice using a Click-iT™ EdU Flow Cytometry Assay Kit (Invitrogen) according to the manufacturer’s instructions.

### 2.5. Multicolor Flow Cytometry Analysis of Tumor-Associated Immune Cells

For flow cytometric analysis, the tumors were dissected into small pieces and subjected to digestion with 40 μg/mL DNase I (Roche, Basel, Switzerland) and 1 mg/mL collagenase D (Roche) for 60 min at 37 °C. Leukocytes were isolated with a 30–70% Percoll gradient (GE Healthcare Life Sciences, Wauwatosa, WI, USA). The spleens of all of the mice were harvested and minced. Bone marrow cells from the femora and tibiae of all mice were flushed with DMEM containing 2% horse serum (Gibco-BRL) and 10 mM HEPES. Single-cell suspensions were washed, and red blood cells (RBCs) were lysed with RBC lysis buffer (BioLegend, San Diego, CA, USA). The cells were resuspended in staining buffer containing 2% horse serum and 0.05% sodium azide, followed by application of an Fc receptor-blocking procedure using anti-mouse CD16/CD32 antibody (clone 2.4G2; BioLegend) for 10 min; they were then immunostained with the following fluorochrome-conjugated primary antibodies: anti-CD45.2 (104), anti-CD3 (145-2C11), anti-CD4 (RM4-5), anti-CD8 (53-6.7), anti-CD11b (M1/70), anti-PD-1 (29F.1A12), anti-TIM3 (RMT3-23), anti-Gr-1 (RB6-8C5), anti-F4/80 (T45-2342), anti-KLRG1 (2F1), anti-IL-7Rα (A7R34), anti-TCF-1/TCF-7 (C63D9), Fixable Viability Stain 780 (BD Biosciences Clontech, Palo Alto, CA, USA), H-2K^b^-SIINFEKL dextramer (Immudex, Copenhagen, Denmark) for the detection of OVA-specific CD8^+^ T cells, and VACV B8R tetramer (H-2K^b^-TSYKFESV tetramer; Immudex) for the detection of VACV-specific CD8^+^ T cells. Following surface staining, the cells were incubated with fixation-permeabilization buffer, washed with permeabilization buffer (Fixation/Permeabilization Solution Kit; BD Biosciences), and incubated with antibodies against intracellular antigens. Samples were processed with a BD Fortessa X-20™ Cell Analyzer (BD Biosciences) and analyzed with FlowJo software (version 10.8.1 FlowJo, Ashland, OR, USA).

### 2.6. Immunofluorescence Microscopy

Immunofluorescence microscopy was performed on the frozen tissue sections. The tumors were fixed in 4% paraformaldehyde at 4 °C and infiltrated with graded sucrose solution (from 15% to 30%). Tissues incubated in 30% sucrose solution were frozen in an optimal cutting temperature compound, and the cut tissues (50-μm-thick) were transferred to a 12-well plate filled with storing solution. Tissue sections were rinsed with phosphate-buffered saline (PBS), blocked in 10% normal goat serum in PBS-T (0.3% Triton X-100 in PBS), and then incubated overnight with the following primary antibodies: anti-cluster of differentiation 31 (CD31)/platelet endothelial cell adhesion molecule-1 (rabbit; Novus Biologicals, Littleton, CO, USA), anti-CD8 (rat, 53-6.7; Novus Biologicals), and anti-CD11b (rabbit; Novus Biologicals). After several washes with PBS, the samples were incubated for 2 h at room temperature with Alexa fluor 594 dye-conjugated anti-rabbit IgG and Alexa-fluor 488 dye-conjugated anti-rat IgG (Abcam, Cambridge, UK). Nuclei were counterstained with 4′,6-diamidino-2-phenylindole (DAPI). Finally, the samples were mounted with fluorescent mounting medium (DakoCytomation Inc., Carpinteria, CA, USA) and images were acquired with an inverted microscope (DMi8; Leica, Wetzlar, Germany).

### 2.7. Statistical Analysis

Statistical analysis was performed using GraphPad Prism (version 8.0; GraphPad Software Inc., San Diego, CA, USA). Group comparisons of tumor growth were carried out by two-way analysis of variance (ANOVA) with Bonferroni correction. For analyzing flow cytometry data, FlowJo software (version 10.8.1; FlowJo) was used. The two-tailed unpaired *t*-test was used to evaluate differences in immunogenicity between groups. Data are expressed as means with standard error of the mean (SEM), and *p* < 0.05 was taken to indicate statistical significance.

## 3. Results

### 3.1. Differential Treatment Efficacy of mJX-594 in Syngeneic Murine Cancer Models

For better evaluation of the immunological characteristics of oncolytic VACV, we analyzed the treatment efficacy of intratumoral administration of mJX-594 using several well-characterized murine cancer cells, including orthotopically transplanted cancers (4T1, EMT6 and B16-F10) and subcutaneously transplanted cancers (LLC, LLC-OVA, MC38, CT26, MB49 and RENCA). Two different doses of mJX-594 (high dose: 5 × 10^7^ pfu, low dose: 2 × 10^7^ pfu) were administered intratumorally, three times at 3-day intervals, starting when the primary tumor mass reached 50–60 mm^3^ (Figure 1A). According to the therapeutic efficacy of mJX-594, murine cancer cells were classified into the following subgroups: more than half of the tumors were in complete remission (CR) after high-dose treatment (EMT6 breast cancer); tumor regression was evident after low-dose treatment and some tumors achieved CR through high-dose treatment (B16-F10 melanoma, RENCA renal cell carcinoma, and MB49 bladder cancer); dose-dependent tumor regression but CR was not achieved through high-dose treatment (LLC non-small-cell lung cancer and MC38 colon cancer); and there was no efficacy of low- or high-dose treatment (CT26 colon cancer and 4T1 breast cancer) (Figure 1, Appendix A).

### 3.2. Marked CD8^+^ T Cell Recruitment in the Tumor following mJX-594 Treatment

To elucidate the differential efficacy and characteristic mechanisms of action of mJX-594 in murine cancer models, we performed multi-color flow cytometry analysis of tumor-infiltrating leukocytes. Both the innate and adaptive arms of immune cells were analyzed 3 days after the last injection of mJX-594 (Figure 2A). The relative percentages of each subset of immune cells among tumor-infiltrating leukocytes (CD45^+^), and the absolute numbers of each subset per milligram of tumor tissue, were determined. All of the murine cancers used in this study contained high proportions (>50%) of immunosuppressive myeloid cells, such as myeloid-derived suppressor cells (MDSCs) and tumor-associated macrophages, and the relative percentages of MDSCs and macrophages differed among cancer cells (Figure 2D,E). T cells were the minority population among tumor-infiltrating leukocytes. Moreover, compared to human cancers, all of the murine cancers in this study were immune desert or immune-excluded phenotypes, with few cells in or around the tumor tissue (Figure 2C,E). At 3 days following the third inoculation of mJX-594, the total CD3^+^ T cell numbers in the tumor tissue increased by at least fourfold, irrespective of therapeutic efficacy in vivo, even with low-dose treatment (Figure 2B and Appendix A). After high-dose treatment, the relative percentage of CD3^+^ T cells among CD45^+^ leukocytes increased by 2.1–7.8 fold, and the cell numbers per milligram of tumor tissue increased by 2.6–26.0 fold (Figure 2B and Appendix A). The increase in intratumoral CD8^+^ T cells was more dramatic. The relative percentage of CD8^+^ T cells among CD45^+^ leukocytes increased by 3.1–19.6 fold, and the cell numbers per milligram of tumor tissue increased by 2.5–72.0 fold (Figure 2B and Appendix A). A marked increase in the CD8^+^ T cell population in the tumor was a key advantage of oncolytic VACV treatment. There was an increase in CD8^+^ T cells in all of the cancers tested, irrespective of clinical efficacy, tissue type, or sensitivity to ICB.

Regarding the immunosuppressive cells recruited by the tumor, each cancer cell had a different immunosuppressive population (typically myeloid cells, such as MDSCs and tumor-associated macrophages, and regulatory T cells) [24]. All of the cancers tested exhibited decreases in the absolute numbers and relative percentages of immunosuppressive myeloid cells after mJX-594 treatment, irrespective of therapeutic efficacy (Figure 2 and Appendix A). A dramatic increase in intratumoral CD8^+^ T cells and a decrease in myeloid cells were confirmed by immunofluorescence microscopy.

Although we cannot accurately predict whether a cancer will respond to mJX-594 therapeutically based on their tumor immune profiles before treatment, we know that mJX-594 treatment markedly increases intratumoral CD3^+^ T cells and (more profoundly) CD8^+^ T cells, and simultaneously decreases immunosuppressive cells such as MDSCs and macrophages in the TME. In summary, oncolytic VACV potently activates and recruits T cells, especially CD8^+^ T cells, in the tumor and renders tumor immune microenvironments more favorable for T cell responses when combined with ICB.

### 3.3. Cancer-Specific CD8^+^ T Cells as Well as Vaccinia Virus-Specific CD8^+^ T Cells Are Efficiently Activated and Recruited to Tumor Tissue

As the administration of VACV equates to an induced viral infection, which triggers effective antiviral cytotoxic T cell responses, we postulated that the increased T cell responses, especially CD8^+^ T cell responses, seen following intratumoral mJX-594 treatment may have included anticancer-specific CD8^+^ T cell responses. To pinpoint the cancer antigen-specific CD8^+^ T cell response, we used the LLC-OVA cancer model, which expresses cytosolic a “whole sequence” OVA protein that serves as a cancer neo-antigen. We evaluated neoantigen (OVA)-specific CD8^+^ T cell responses using the K^b^-SIINFEKL dextramer. The VACV-specific CD8^+^ T cell response was evaluated using the K^b^-TSYKFESV tetramer, which detected an epitope from B8R encoding a secreted protein with homology to the interferon gamma receptor [25]. mJX-594 (5 × 10^7^ pfu) was administered intratumorally, three times at 3-day intervals, starting when the primary LLC-OVA tumor reached 50–60 mm^3^ (Figure 1A). Tumor-infiltrating lymphocytes were analyzed on day 3 after the last virus administration (i.e., 9 days after the first virus treatment), during which the CD8^+^ T cell response was maximal. mJX-594 treatment increased OVA-specific CD8^+^ T cell expression, as well as VACV-specific CD8^+^ T cell expression, more than 10-fold in terms of the percentage and 30–50-fold in terms of the number of cells per milligram of tumor tissue, compared to vehicle treatment (Figure 3A–C). These results show that mJX-594 effectively infected and destroyed cancer cells, and released cancer antigens in an immunogenic manner, thus efficiently activating and recruiting cancer antigen-specific CD8^+^ T cells into the tumor tissue.

### 3.4. Anti-PD-1 Combination Therapy Increases Treatment Efficacy and Intratumoral Infiltration of T Cells in Two Murine Cancer Models

As mJX-594 markedly increased intratumoral CD8^+^ T cells in all syngeneic murine cancer cells tested irrespective of their therapeutic efficacy, mouse strain, or tumor immune microenvironment before inoculation, and also increased cancer antigen-specific CD8^+^ T cells in the tumor, we considered mJX-594 to be among the most powerful reagents for converting immune desert or immune-excluded cancers into inflamed ones, thus making them susceptible to ICB therapies, such as anti-PD-1 and anti-programmed death-ligand 1 (PD-L1) therapies. We tested 4T1 cancer cells, which are resistant to anti-PD-1/anti-PD-L1 antibody therapy [24], and found that they did not respond to mJX-594 monotherapy at a high dose, but nevertheless exhibited an increase of intratumoral CD8^+^ T cells (by 6.7-fold) following mJX-594 treatment (Figure 2E and Appendix A). Single therapy, of either high-dose mJX-594 or anti-PD-1 antibody, did not have significant tumor-reducing effects in 4T1-bearing mice. Combined treatment with mJX-594 and anti-PD-1 had anti-cancer effects on the murine triple-negative breast cancer 4T1 (Appendix A). These results showed that mJX-594 treatment sensitizes one of the most resistant murine cancer cells to anti-PD-1 antibody therapy.

To confirm the synergistic effect of mJX-594 and anti-PD-1 antibody, we used the MC38 murine colon cancer model, and combined low-dose mJX-594 and anti-PD-1 antibody therapy. Low-dose mJX-594 single therapy achieved 37% inhibition, while anti-PD-1 single therapy did not achieve a significant reduction. Combination therapy achieved tumor growth inhibition of 69% (Figure 4A–C). Increased numbers of CD8^+^ T cells in the tumor tissue, as induced by mJX-594 treatment, were confirmed by immunofluorescence microscopy on days 9 and 15 of the first mJX-594 injection (Figure 4A). Vehicle treatment was associated with few CD8^+^ T cells in tumor tissue, and anti-PD-1 antibody single therapy slightly increased CD8^+^ T cell expression. However, single therapy of mJX-594 dramatically increased CD8^+^ T cell infiltration into the tumor tissue, and combined anti-PD-1 antibody further increased these cell populations (Figure 4D,E). A dramatic reduction of the CD11b^+^ myeloid population, the majority of which comprised MDSCs (Figure 2E), by mJX-594 single therapy and combination therapy with anti-PD-1 antibody was observed and anti-PD-1 single therapy did not significantly decrease these cell populations (Figure 4D,E). Reduction of the CD31 area (Figure 4D,E) reflected the well-known tumor-associated endothelial cell damage induced by VACV [26].

### 3.5. Among Cancer Antigen-Specific T Cells, TCF1^+^ Stem-like CD8^+^ T Cells Are Increased by mJX-594 and Further Increased by Anti-PD-1 Antibody Combination Treatment

As the production of TCF1^+^ stem-like CD8^+^ T cells is related to protective immunity following cancer vaccination [16], the prognosis of the patient [12], and putative future memory T cells [27], we evaluated whether stem-like CD8^+^ T cells were increased by mJX-594 treatment. We inoculated LLC-OVA cancer cells into the subcutaneous region and applied 5 × 10^7^ Pfu mJX-594 intratumorally three times at 3-day intervals, starting when the primary tumor mass reached 50–60 mm^3^. Anti-PD-1 antibody was intravenously injected, starting on day 3 of mJX-594 inoculation, at 3-day intervals a total of three times. Tumor growth was inhibited by 55% by high-dose treatment of mJX-594 but was not inhibited by anti-PD-1 antibody monotherapy. A synergistic tumor-inhibiting effect of the combined treatment was also seen (Figure 5A).

Tumor-infiltrating leukocytes and splenocytes were analyzed 3 days after the last injection of mJX-594 (day 9 after the first injection). The mJX-594 treatment was highly effective for CD8^+^ T cell differentiation. KLRG1^+^ CD8^+^ T cells were markedly increased by mJX-594 treatment and further increased by anti-PD-1 combination treatment in both the tumor and spleen (Figure 5B–D). Anti-PD-1 monotherapy did not appreciably increase effector cell expression. Both KLRG1^+^IL-7Rα^−^ short-lived effector cells and KLRG1^+^IL-7Rα^+^ memory precursor effector cells were increased (Figure 5B,C). GZMB^+^ CD8^+^ T cells were also prominently increased by mJX-594 treatment and further increased by anti-PD-1 combination treatment in the spleen (Figure 5E,F).

TCF-1^+^PD-1^+^ stem-like CD8^+^ T cells were also increased by mJX-594 treatment in the tumor and the spleen 9 days after the injection and further increased by anti-PD-1 combination treatment in the spleen (Figure 5H–J). On day 30 following the last injection (36 days after the first injection), increased percentages of TCF-1^+^PD-1^+^ stem-like CD8^+^ T cells were noted in both the spleen and bone marrow (Figure 5K–N).

K^b^-SIINFEKL^+^ OVA-specific CD8^+^ T cells in the tumor were increased by mJX-594 and further increased by combined treatment involving anti-PD-1 (Appendix A). Among K^b^-SIINFEKL^+^ CD8^+^ T cells, KLRG1^+^ effector T cells were also increased in both the tumor and spleen by mJX-594 treatment (Appendix A). TCF-1^+^PD-1^+^ stem-like K^b^-SIINFEKL^+^ CD8^+^ T cells were also increased by mJX-594 treatment in the tumor and spleen on day 9 (Appendix A). On day 36 after the first injection, increased percentages of TCF-1^+^PD-1^+^ stem-like K^b^-SIINFEKL^+^ CD8^+^ T cells were noted in both the spleen and bone marrow (Appendix A).

### 3.6. Cancer Neoantigen-Specific CD8^+^ T Cells That Survive for Extended Periods in the Spleen and Bone Marrow Proliferate in Response to Antigen-Loaded DCs In Situ

We showed that TCF-1^+^PD-1^+^ stem-like CD8^+^ T cell expression was increased in terms of both total and cancer neoantigen-specific CD8^+^ cells on day 9 after the first viral infection, and it remained elevated on day 36. To confirm the functional capacity of the cancer-specific stem-like/memory-like T cells in the lymphoid organs in situ following mJX-594-mediated cancer treatment, we utilized the LLC-OVA cancer model. We subcutaneously inoculated LLC-OVA cancer cells into mice, followed by intratumoral treatment with 5 × 10^7^ pfu of mJX-594 (five times at 3-day intervals starting when the primary tumor mass reached 50–60 mm^3^). To detect functional stem-like/memory-like T cells in specific organs, we used antigen-loaded splenic DCs for in situ activation of local antigen-specific T cells. As systematically injected DCs localize in the bone marrow and secondary lymphoid organs [28], we loaded SIINFEKL peptide (OVA peptides 357–364) onto splenic DCs, which were then injected intravenously. Proliferating K^b^-SIINFEKL-specific CD8^+^ T cells were evaluated using a CD45 congenic marker, K^b^-SIINFEKL dextramer, and the EdU click chemistry method [29] on days 39 and 60 (Figure 6) after the first mJX-594 injection. We found that cancer neoantigen-specific T cells in the spleen and bone marrow proliferated in response to SIINFEKL-loaded splenic DCs (Figure 6). These results indicated that cancer neoantigen-specific T cells were present in the secondary lymphoid organs and bone marrow for a protracted period and were also functional (as they responded to antigen-loaded splenic DCs).

## 4. Discussion

In this study, we evaluated the immune-modulating characteristics of oncolytic VACV using murine syngeneic cancer models. We found that oncolytic VACV Pexa-vec (mJX-594) potently increased CD8^+^ T cells, including cancer antigen-specific CD8^+^ T cells, and decreased immunosuppressive cells in tumors irrespective of therapeutic efficacy (Figure 2). These findings indicate that mJX-594 was highly effective at transforming immune desert and immune-excluded tumors into inflamed ones and modulating the immunosuppressive TME. These typical assets of mJX-594 were ideal for combination therapy involving anti-PD-1/anti-PD-L1, which only targets inflamed tumors in which treatment reinvigorated immune exhausted T cells in the TME [30]. Indeed, mJX-594 led to a response to combination anti-PD-1 blocking antibody treatment in one of the most resistant murine cancer cell lines, which did not respond to mJX-594 single therapy or anti-PD-1 single therapy (Appendix A) [24].

In the TME and cancer-bearing hosts, the functional capacity of cancer-specific effector CD8^+^ T cells was compromised by the expression of several negative immune checkpoint molecules on their surfaces, which led to their exhaustion [8,9]. Among these dysfunctional cancer-infiltrating T cells, which are quite similar to the exhausted cells seen in chronic viral infection, some subpopulations were still functional, where the number of these subpopulations correlated with the overall survival of several cancer types of human cancers [12,14]. The cells in these subpopulations proliferated following anti-PD-1 therapy and differentiated into effector T cells that controlled the tumor and maintained homeostasis, thus preventing further tumor progression [12,31]. The phenotype of these cells was TCF1^+^ PD-1^+^, which are referred to as either stem-like or progenitor-exhausted T cells [32]. The proliferation of stem-like T cells is positively correlated with the clinical outcomes and overall survival of patients [12,14,33], and a recent study showed that the efficacy of neoantigen cancer vaccines was correlated with stem-like T cell generation [16]. In this study, mJX-594 treatment was quite efficient for increasing cancer antigen-specific stem-like CD8^+^ T cells in the tumor tissue and secondary lymphoid tissue, such as the spleen and bone marrow. Moreover, the quantity of these stem-like T cells correlated with the therapeutic efficacy of mJX-594 and was further increased by (and correlated with) combination therapy involving anti-PD-1 (Figure 5 and Appendix A). Thus, mJX-594 acted as an antigen-agnostic vaccine platform, which could activate and expand the population of cancer-specific stem-like CD8^+^ T cells in association with the therapeutic efficacy of this treatment modality.

In acute viral infection, TCF1^+^ T cells expressed following infection serve as early memory precursor cells, which eventually differentiate into permanent memory T cells [27]. Memory T cell generation is the ultimate goal of cancer immunotherapy and therapeutic cancer vaccines for the prevention of recurrence of cancer in the future [34]. However, as in chronic viral infection, this ordinary differentiation process involving the formation of permanent memory T cells following activation is compromised in cancer [34,35]. In this study, mJX-594 treatment helped normalize the TME, which allowed for differentiation of functional effector and TCF1^+^ stem-like CD8^+^ T cells. In addition, we observed an increase in the expression of cancer-specific TCF1^+^ stem-like CD8^+^ T cells following mJX-594 treatment of the spleen and bone marrow for an extended period. In murine and human tumor models, stem-like T cells were found in the TME [12,14,31], and some reports have indicated that tertiary lymphoid structures within the tumor tissue are the main sites for these T cells [36]. In chronic viral infection, stem-like CD8^+^ T cells are evident in the secondary lymphoid organs and respond to anti-PD-1 antibody by proliferating and differentiating into effector cells [11,37]. Regarding the location of permanent memory T cells, the bone marrow and secondary lymphoid organs are the sites of the central memory T cells, whereas the target tissues are effector memory T cells and the recently characterized resident memory T cells [38,39,40]. The existence of cancer-specific stem-like CD8^+^ T cells in the secondary lymphoid organs noted in this study is relevant to these previous reports, and in terms of long-term survival of functional cancer antigen-specific T cells in the spleen and bone marrow, which proliferated upon encountering cancer antigen-loaded splenic DCs 2 months after cancer treatment (Figure 6), indicating that mJX-594 treatment elicits long-term anticancer immune responses.

## 5. Conclusions

Intratumorally administered oncolytic VACV (mJX-594, a murine variant of JX-594, Pexa-vec) potently increased CD8^+^ T cell proliferation, including of cancer antigen-specific CD8^+^ T cells, and decreased immunosuppressive cells irrespective of tissue type or therapeutic efficacy. Remodeling of tumors into inflamed ones by mJX-594 led to a response to combined anti-PD-1 treatment but not to mJX-594 or anti-PD-1 monotherapy. Among cancer-specific CD8^+^ T cells, mJX-594 treatment increased the expression of TCF1^+^ stem-like T cells, while anti-PD-1 combination treatment further increased their expression, which was important for therapeutic efficacy. The presence of functional cancer-specific CD8^+^ T cells in the spleen and bone marrow for an extended period, where these cells proliferated upon encountering cancer antigen-loaded splenic DCs, further indicates that long-term anti-cancer immunity is elicited by oncolytic VACV.

## Figures and Tables

**Figure 1 biomedicines-10-00805-f001:**
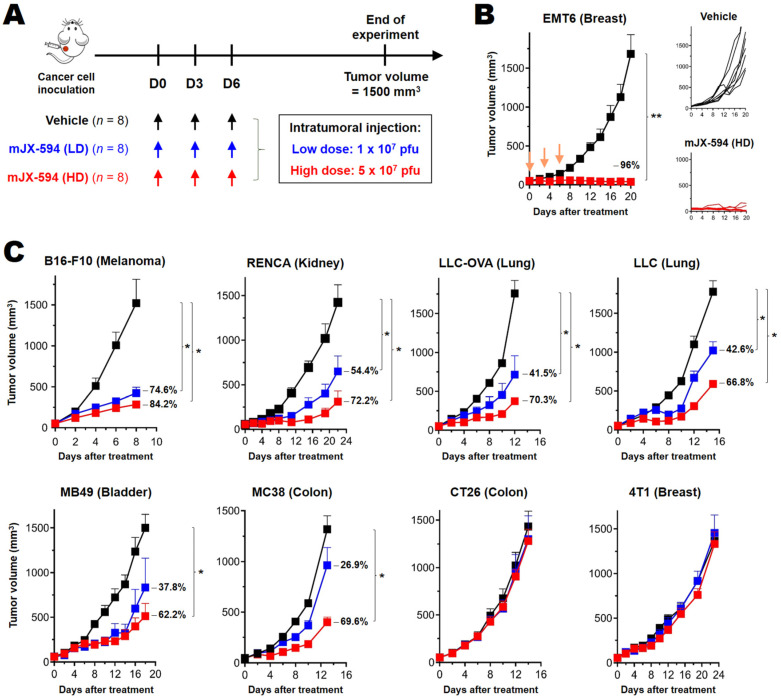
Differential treatment efficacy of mJX-594 in syngeneic murine cancer models. (**A**) mJX-594 treatment strategy. Cancer cells were implanted subcutaneously into C57BL/6 and BALB/c mice (*n* = 8 per group). When the tumor reached a volume of 50–60 mm^3^, mice were intratumorally injected, three times at 3-day intervals, with vehicle, 1 × 10^7^ pfu of mJX-594 (LD), or 5 × 10^7^ pfu of mJX-594 (HD). (**B**) Tumor growth kinetics of EMT6 tumors treated with vehicle or mJX-594 (HD). (**C**) Tumor growth kinetics of B16-F10, RENCA, LLC-OVA, LLC, MB49, MC38, CT26, and 4T1 tumors treated with vehicle, mJX-594 (LD), or mJX-594 (HD). The inhibitory effects (%) in the LD and HD groups were compared to the vehicle control group. Group comparisons of tumor growth were carried out by two-way analysis of variance (ANOVA) with Bonferroni correction. Data are expressed as means with standard error of the mean (SEM). * *p* < 0.05, ** *p* < 0.005. LLC-OVA, LLC expressing OVA; HD, high dose; LD, low dose; OVA, ovalbumin; pfu, plaque-forming units.

**Figure 2 biomedicines-10-00805-f002:**
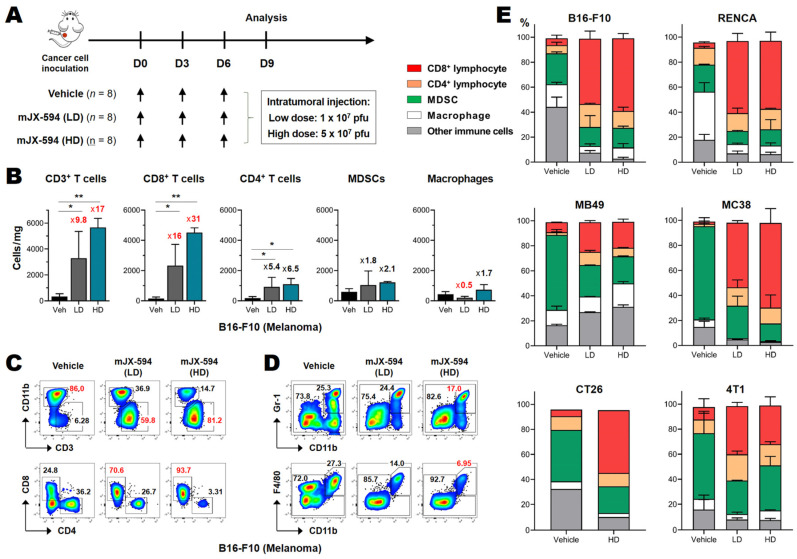
Profound CD8^+^ T cell activation and decreased expression of suppressive myeloid cells by mJX-594. (**A**) mJX-594 treatment strategy. Cancer cells were implanted subcutaneously into C57BL/6 and BALB/c mice. When the tumor reached a volume of 50–60 mm^3^, mice were intratumorally injected, three times at 3-day intervals, with vehicle, 1 × 10^7^ pfu of mJX-594 (LD), or 5 × 10^7^ pfu of mJX-594 (HD). The tumor-infiltrating leukocytes were analyzed 3 days after the last treatment with mJX-594. (**B**) The numbers of CD3^+^ T cells, CD8^+^ T cells, CD4^+^ T cells, MDSCs, and macrophages per milligram of tumor mass among infiltrated leukocytes of B16-F10 melanoma. (**C**) Flowcytometric analysis and gating strategies for CD3^+^ T cells and CD11b^+^ myeloid cells among infiltrated leukocytes (upper) and CD8^+^ T cells and CD4^+^ T cells among infiltrated CD3^+^ cells (lower) of B16-F10 melanoma. (**D**) Flowcytometric analysis and gating strategies for MDSCs (CD11b^+^ Gr-1^+^) (upper) and macrophages (F4/80^+^ CD11b^+^) (lower) among infiltrated leukocytes of B16-F10 melanoma. (**E**) Summarized relative proportions of CD8^+^ lymphocyte, CD4^+^ lymphocyte, MDSC, macrophage, and other immune cells among infiltrated leukocytes of B16-F10, RENCA, MB49, MC38, CT26, and 4T1 tumors. Pooled data from two experiments are shown (*n* = 6 per group). * *p* < 0.05, ** *p* < 0.005. MDSC, myeloid-derived suppressor cells; mJX, mJX-594; OVA, ovalbumin; pfu, plaque-forming units; Veh, vehicle.

**Figure 3 biomedicines-10-00805-f003:**
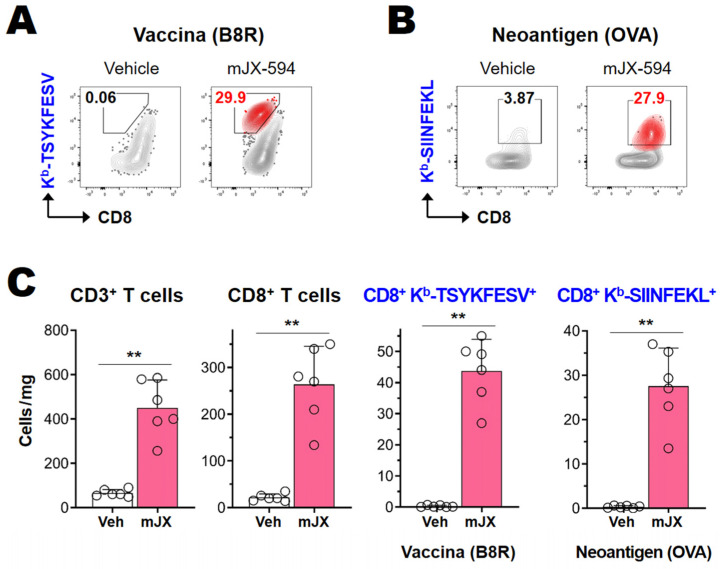
Cancer-specific CD8^+^ T cells and vaccinia virus-specific CD8^+^ T cells were efficiently activated and recruited to the tumor tissue. LLC-OVA cancer cells were subcutaneously implanted into C57BL/6 mice. When tumors reached 50–60 mm^3^ in volume, the mice were intratumorally injected with vehicle or 5 × 10^7^ pfu of mJX-594 on days 0, 3, and 6. The tumor-infiltrating leukocytes were analyzed by flow cytometry on day 3 after the last treatment with mJX-594. (**A**) The proportion of vaccinia virus B8R-specific CD8^+^ T cells (K^b^-TSYKFESV tetramer-positive) among CD3^+^ T cells. (**B**) The proportion of neoantigen (OVA)-specific T cells (K^b^-SIINFEKL dextramer-positive) among CD3^+^ T cells. (**C**) Cell number per milligram of tumor mass of intratumoral CD3^+^ T cells, CD8^+^ T cells, B8R vaccinia-specific CD8^+^ T cells, and OVA-specific CD8^+^ T cells after mJX-594 treatment. Pooled data from two experiments are shown (*n* = 6 per group). ** *p* < 0.005. LLC-OVA, LLC expressing OVA; mJX, mJX-594; OVA, ovalbumin; pfu, plaque-forming units; Veh, vehicle.

**Figure 4 biomedicines-10-00805-f004:**
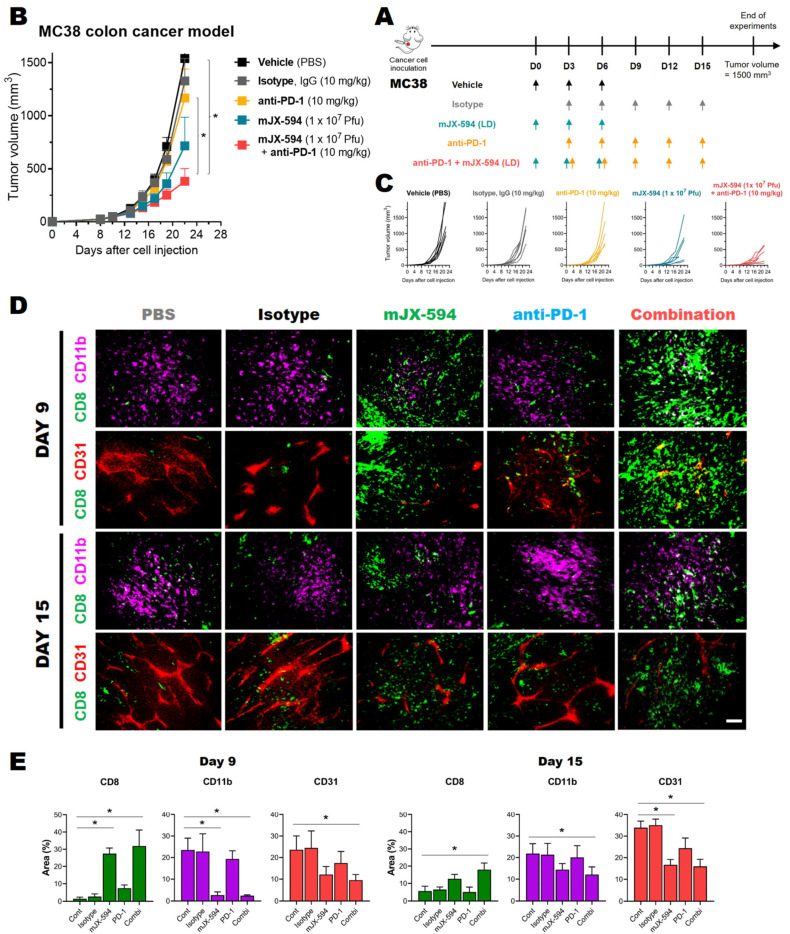
Combination treatment of anti-PD-1 with mJX-594 increased anti-tumor effects in a MC38 colon cancer model. (**A**) Combination treatment strategy. MC38 cancer cells were subcutaneously implanted into C57BL/6 mice. When tumors reached 50–60 mm^3^ in volume, mice were intratumorally administered 1 × 10^7^ pfu of mJX-594 on days 0, 3, and 6. For combination therapy, anti-PD-1 antibody was intravenously injected five times at 3-day intervals, starting from the second treatment of mJX-594. Tumor growth monitoring until the end of the experiment (tumor volume ≤ 1500 mm^3^), and immunofluorescence microscopy images showing changes in immune cell infiltration on days 9 and 15 after the first mJX-594 treatment in MC38 tumor tissue. (**B**) Average tumor growth curve of MC38 tumor-bearing mice after mJX-594 and anti-PD-1 combination treatment (*n* = 7 per group). Group comparisons of tumor growth were carried out by two-way ANOVA with Bonferroni correction. Values are mean ± SEM. * *p* < 0.05. (**C**) Individual tumor growth curves of MC38 tumor bearing mice after mJX-594 and anti-PD-1 combination treatment. (*n* = 7 per group) (**D**) Representative images of CD8^+^ T cells, CD31^+^ blood vessels, CD11b^+^ myeloid cells in tissue sections. Scale bars, 200 mm. (**E**) Percentage of area infiltrated with leukocytes relative to the whole tumor, calculated using ImageJ software on days 9 and 15 after the first mJX-594 treatment. * *p* < 0.05. Two-tailed Student’s *t*-test was used for the analysis.

**Figure 5 biomedicines-10-00805-f005:**
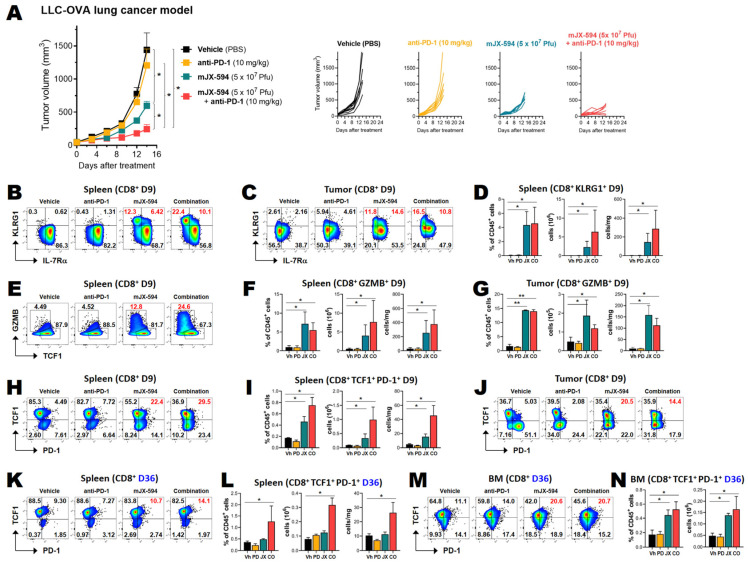
Tcf1^+^ stem-like CD8^+^ T cells including those directed against tumor-associated antigens, were increased by mJX-594 and further increased by anti-PD-1 antibody combination. LLC-OVA cancer cells were subcutaneously implanted into C57BL/6 mice. When tumors reached 50–60 mm^3^ in volume, the mice were intratumorally injected with vehicle or 5 × 10^7^ pfu of mJX-594 on days 0, 3, and 6. For combination therapy, anti-PD-1 antibody was intravenously injected five times at 3-day intervals, starting from the second treatment of mJX-594. Tumor growth was monitored until the end of the experiment (tumor volume ≤ 1500 mm^3^). The tumor-infiltrating leukocytes and splenocytes were analyzed by flow cytometry at 9 days (D9), and splenocytes and bone marrow cells at 36 days (D36), after the first treatment with mJX-594. (**A**) Average tumor growth curve (**left**) and individual tumor growth curves (**right**) of LLC-OVA tumor-bearing mice after mJX-594 and anti-PD-1 combination treatment (*n* = 7 per group). Group comparisons of tumor growth were carried out by two-way ANOVA with Bonferroni correction. Values are mean ± SEM. * *p* < 0.05. (**C**) Individual tumor growth curves of MC38 tumor-bearing mice after mJX-594 and anti-PD-1 combination treatment. (*n* = 7 per group). (**B**,**C**) Relative proportions of memory precursor effector cells (IL-7Rα^+^KLRG1^int^) and short-lived effector cells (IL-7Rα^−^KLRG1^high^) among CD8^+^ T cells in spleen and tumor (D9). (**D**) Relative proportion among infiltrating leukocytes, the total cell numbers, and the number of cells per unit gram of tissue of KLRG1^+^ effector CD8^+^ T cells in spleen (D9). (**E**) Relative proportion of effector cells (TCF1^−^GzmB^+^) among CD8^+^ T cells in spleen (D9). (**F**,**G**) GzmB^+^ effector CD8^+^ T cells in spleen and tumor (D9). (**H**,**I**) TCF1^+^PD-1^+^ stem-like CD8^+^ T cells in spleen (D9). (**J**) TCF1^+^PD-1^+^ stem-like CD8^+^ T cells in tumor (D9). (**K**,**L**) TCF1^+^PD-1^+^ stem-like CD8^+^ T cells in spleen (D36). (**M**,**N**) TCF1^+^PD-1^+^ stem-like CD8^+^ T cells in tumor (D36). Pooled data from two experiments are shown (*n* = 6 per group). * *p* < 0.05, ** *p* < 0.005. CO, combination therapy; GZMB, granzyme B; LLC-OVA, LLC expressing OVA; JX, mJX-594; OVA, ovalbumin; PD, anti-PD-1; pfu, plaque-forming units; Vh, vehicle.

**Figure 6 biomedicines-10-00805-f006:**
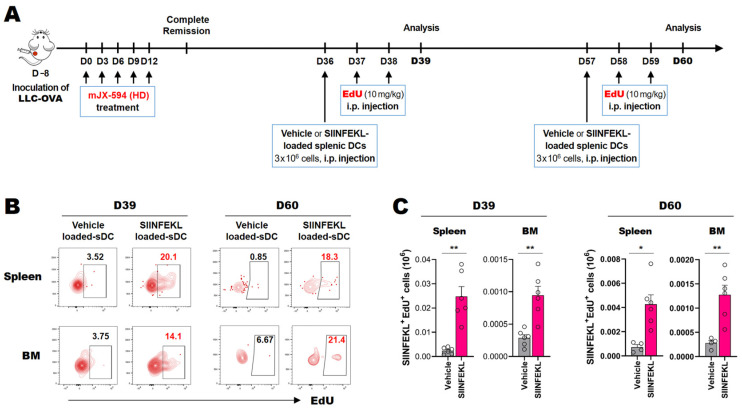
Long-lived cancer neoantigen-specific CD8^+^ T cells proliferated in response to antigen-loaded dendritic cells in situ. (**A**) Schematic representation of the experimental schedule. LLC-OVA cells were subcutaneously implanted into C57BL/6 mice. When the tumor volume reached 50–60 mm^3^, the mice were intratumorally injected with 5 × 10^7^ pfu mJX-594 five times at 3-day intervals. Intraperitoneal injection of 3 × 10^6^ splenic dendritic cells (sDCs) that had been loaded with either vehicle or 100 nM SIINFEKL on day 36 or day 57 of the first mJX-594 injection. In vivo proliferation of cancer neoantigen-specific memory-like CD8^+^ T cells in response to antigen-loaded sDCs was measured by incorporation of EdU (10 mg/kg) after labeling the cells twice (once per day on day 1 and day 2 before analysis) and analyzed by K^b^-SIINFEKL dextramer staining and Click-iT™ EdU Flow Cytometry Assay Kit (K^b^-SIINFEKL^+^ EdU^+^). (**B**,**C**) Representative flowcytometric plot (**B**) and absolute number (**C**) of proliferating EdU^+^ cells among K^b^-SIINFEKL dextramer-positive cells. Pooled data from two experiments are shown (*n* = 4–6 per group). * *p* < 0.05, ** *p* < 0.005. BM, bone marrow; EdU, 5-ethynyl-2′-deoxyuridine; i.p., intraperitoneal; LLC-OVA, LLC expressing OVA; OVA, ovalbumin; pfu, plaque-forming units; sDC, splenic dendritic cells.

## Data Availability

Data are available on reasonable request. All data relevant to the study are included in the article or uploaded as Appendix A.

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
