# Peer review of "Oncolytic Vaccinia Virus Augments T Cell Factor 1-Positive Stem-like CD8+ T Cells, Which Underlies the Efficacy of Anti-PD-1 Combination Immunotherapy"

_biomedicines, 2022, doi:10.3390/biomedicines10040805_

Round 1

Reviewer 1 Report

The manuscript “Oncolytic vaccinia virus augments T cell factor 1-positive stem-like CD8+T cells, which underlies the efficacy of anti-PD-1 combination immunotherapy” is a comprehensive study of the oncolytic effect of vaccinia virus associated with immune aspects. The authors carried out a large complex of complementary studies, built into a logical chain of studying the antitumor effect of oncolytic vaccinia virus on different variants of syngenic tumors in mice, focusing on T8 lymphocytes, which serve as the main killer cells of tumor cells in the body. The "Results" section is well structured, the data obtained are quite fully illustrated.

However, it should be noted that the text of the article is difficult to read, mostly, due to the abundance of abbreviations, not all of which are deciphered and listed in the list of abbreviations. Considering that not all potential readers can easily recognize this or that abbreviation, I recommend that the authors carefully check this point and at least compile a complete list of abbreviations. For the same purpose, to "facilitate" the reader to read and understand the article, I recommend carefully editing the text and figure captions to make them as understandable as possible.

Some minor comments to “Materials and methods”.

In section “Cell lines”, please, provide complete names of listed cell lines indicating their origination (EMT6, B16-F10, RENCA, LLC, MB49, MC38, CT26, and 4T1). This is general requirements and such information is useful for somebody wanting to compare your data with published or obtained.

Section “Oncolytic virus”

Additional information is needed to clarify the experiments.

It is clear from the text that the virus preparations were obtained from SillaJen company. However, it is necessary to provide such information as:

  • what substrate was used to produce the virus (cell line, which one? chicken embryos?);
  • what was pretreatment of the virus preparation (purification, concentration)

Section “Tumor models and treatment regimens”

Please provide data on the volume of the injected virus.

In a total, the work presented is interesting and shows new possibilities in development of cancer treatment, as well as a new step in development of treatment methods based on oncolytic viruses.  Obviously, the mouse model does not reproduce the disease in humans, however, this work demonstrates the capabilities of oncolytic viruses, taking into account their general biological properties.

Reviewer 2 Report

The authors have shown the efficacy of the combination of JX-594 and anti-PD-1 using many mouse syngeneic cancer models. The combination also showed changes in the proportion of immune cells, including TCF1+ PD-1+ CD8+ T cells in TME, spleen, and BM.  

No supplementary data can be found in this manuscript.

The authors focused on the immune response from JX-594 infection. They showed that T26 and 4T1 cells were resistant to the antitumor effect of JX-594 (Figure 1). Can JX-594 infect CT26 and 4T1 cells in vitro, express viral antigens and induce cytopathic effects? Do these cells express PD-L1 and are affected by anti-PD-1?

Figure 2: The authors examined the proportion of TCF1+ stem-like CD8+ T cells in MC38 (Figure 4) and LLC-OVA cells (Figure 5). The main purpose of this study is to demonstrate the changes in various immune cells that infiltrate TME with JX-594 treatment. Why are the results of LLC or LLC-OVA cells not shown in Figure 2?

Figure 4: The authors show that anti-PD-1 alone does not significantly affect the antitumor effect and the proportion of immune cells in MC38 tumors, but it enhanced the ability of JX-594. Did the authors confirm that JX-594 enhanced PD-L1 expression on tumor cells?

Figure 4 and figure 5: Combination therapy has the most potent effect on the tumor growth and expansion of CD8+ cells, including TCF1+ PD-1+ CD8+ T cells. However, the difference between the JX-594 group and the JX-594+anti-PD-1 group did not reach the significant level. Therefore, it is difficult to conclude that anti-PD-1 clearly enhanced the antitumor capability and CD8+ T cells expanding effect of JX-594. This needs to be explained in the results and discussion.

Figure 5J: In combination therapy, the population of TCF1+PD-1+ CD8+ T cells increased in the spleen and BM. However, the stem-like CD8+ T cell population did not increase in tumors as in spleen and BM. In this regard, Jansen et al. stated that stem-like CD8+ T cells reside in dense antigen-presenting-cell niches within the tumor, and that tumors that fail to form these structures are not extensively infiltrated by T cells (ref 14). It is necessary to explain the difference between tumor and spleen/BM as the site of stem-like CD8+ T cell accumulation.

The “JX-594” described in the abstract is incorrect. Instead, the authors need to explain mJX-594 and what it means.

Kb-TSYKFESV tetramer should be explained in Materials and Methods.

Reviewer 3 Report

The authors used the vaccinia virus-based oncolytic virus (mJX-594) for this preclinical study.  They also used many immunocompetent mouse models to evaluate therapeutic efficacy of mJX-594).

However, mJX-594 showed differential therapeutic efficacy in various cancer models. To investigate this, infiltrated immune cells in tumor microenvironment were also quantified. The results showed differential populations of immune cells in TME.

Furthermore, to differentiate infiltrations of cancer-specific CD8+ T cells and virus-specific CD8+ T cells in TME, the LLC-OVA model was used in this study.

  α-PD-1 was used for combination therapy. In MC-38 cancer model, low and high doses of mJX-594 were used with α-PD-1. TCF1+ stem-like CD8+ T cells infiltration was also shown in LLC-OVA models.

Reviewer 4 Report

This is a well written manuscript reporting the ability of a vaccinia virus recombinant encoding murine GMCSF (mJX-574) to interfere with propagation of a variety of syngeneic murine cancer cell lines grafted into mice. The authors demonstrate that inoculation of the oncolytic virus into tumours at an early stage of development leads to infiltration of CD8+ T cells and the decrease within the tumours of suppressive myeloid cells. Nevertheless, the authors found that there is no direct correlation between protection against tumour growth, which does not occur for some cancer cell lines, and infiltration of T cells which occurs for all of them. The authors go on to show that antigen specific CD8+ T cells found within the tumours are directed both against tumour antigens and vaccinia virus antigens. In the case of one murine cancer cell line they show that infection with the oncolytic virus combined with treatment with an anti-PD1 antibody provides efficient control of tumour growth that was not observed with either viral infection or anti-PD1 treatment alone. Moreover, they demonstrate that TCF1+ stem-like CD8+ T cells are particularly amplified by the combined treatment suggesting that these cells are the critical players in providing protection against tumour growth at least for one cancer cell line. Finally, the authors have performed experiments indicating that immunity generated by the oncolytic virus can be relatively long-lived within the animals. Overall the data presented are convincing and they support the conclusions drawn. I have only a few comments on details of the manuscript.

1.To some extent the title of the manuscript is misleading since it puts emphasises on the findings for one tumour cell line while ignoring that combined immunotherapy was not required to prevent growth of several of the other cancer cell lines injected and not even investigated for most of them. A title that better reflects the entire study would be more appropriate.  

2.The abstract as well as the manuscript should refer to mJX-594 throughout and not JX-594 because the former and not the latter has been used in this study. They of course could briefly justify use of mJX-594 in the abstract and the manuscript (as already done in the latter case).

  1. VACV and not VacV is the more common abbreviation of vaccinia virus.
  2. It might be useful for some readers that the origin of the murine cancer cell lines (either C57BL/6 or BALB/c) be mentioned in the methods section.
  3. The sentence at the bottom of page 13 (The fold differences in CD8+ T cell increases did not correspond to the VacV anticancer efficacy) is actually a repetition of the sentence immediately before it.
  4. page 18 line 9 refers to Figure 1C and should actually refer to figure 2E.
